# Synovial Glucose and Serum-to-Synovial Glucose Predict Failure After Acute Postoperative Infection in Total Knee Arthroplasty

**DOI:** 10.3390/jcm14082841

**Published:** 2025-04-20

**Authors:** Marta Sabater-Martos, Laia Boadas, Laura Morata, Alex Soriano, Juan Carlos Martínez-Pastor

**Affiliations:** 1Orthopedic and Traumatology Department, Clínic Barcelona, Carrer Villarroel 170, 08036 Barcelona, Spain; laiaboadas93@gmail.com (L.B.); jcmartin@clinic.cat (J.C.M.-P.); 2Department of Infectious Diseases, Clínic Barcelona, University of Barcelona, IDIBAPS (Institut d’Investigacions Biomèdiques Agustí-Pi Sunyer), Carrer Villarroel 170, 08036 Barcelona, Spain; lmorata@clinic.cat; 3CIBERINF, CIBER in Infectious Diseases, 28029 Madrid, Spain; asoriano@clinic.cat

**Keywords:** PJI, DAIR, risk factor, synovial fluid

## Abstract

**Background**: The treatment of periprosthetic joint infection (PJI) involves various strategies, with debridement, antibiotic, and implant retention (DAIR) being a preferred method for acute infections due to its lower morbidity. However, DAIR success rates vary widely from 30% to 80%. This study investigates the predictive value of synovial glucose and the serum-to-synovial glucose ratio for DAIR outcomes in acute postoperative PJI following total knee arthroplasty (TKA). **Methods**: This is a retrospective study of 32 DAIR cases, diagnosed with acute postoperative PJI after TKA. Synovial joint aspirations were performed on all patients. We collected all serological and synovial glucose levels. The serum-to-synovial glucose ratio was calculated. **Results**: Patients with synovial glucose levels below 44 mg/dL and a serum-to-synovial glucose ratio above 50% were identified as high risk for DAIR failure. High-risk patients exhibited a 31.3% failure rate, with half occurring within the first three months post-DAIR. No failures were observed in the low-risk group. Multivariate analysis did not find other significant predictors such as CRP levels, gender, or microbial cultures. **Conclusions**: Low synovial glucose levels and high serum-to-synovial glucose ratios are predictive of unsuccessful outcomes following DAIR procedures. Patients exhibiting lower synovial concentrations experienced early treatment failure.

## 1. Introduction

Periprosthetic joint infection (PJI) remains one of the most challenging and devastating complications following total joint arthroplasty. Despite advances in preventative strategies and surgical techniques, the incidence of PJI has remained relatively constant, with significant implications for both patient morbidity and healthcare costs [1,2,3,4,5]. In the early postoperative setting, acute infections may be managed with debridement, antibiotics, and implant retention (DAIR), a treatment strategy aimed at preserving the prosthesis while controlling the infection. DAIR is considered one of the surgical options with the lowest associated morbidity [6,7], making it an attractive alternative to more invasive revision procedures.

However, the success rate of DAIR is highly variable, ranging from 30% to 80% depending on patient-related factors, microbial virulence, surgical technique, and timing of intervention [8,9,10,11]. This variability underscores the urgent need for reliable predictors of treatment outcomes, which would aid in clinical decision-making and potentially improve patient selection for DAIR. Several prognostic tools have been developed to help predict DAIR failure, including a score specifically designed for acute postoperative PJI [12] and another tailored for late acute hematogenous PJI [13]. More recently, Shohat et al. [14] proposed a multifactorial model incorporating ten clinical and laboratory parameters, identifying C-reactive protein (CRP) as the most significant biomarker associated with failure.

In the context of native septic arthritis, synovial fluid analysis—particularly glucose levels—has long been established as a useful diagnostic tool [15,16]. A marked decrease in synovial glucose concentration is typically observed in joint infections due to increased metabolic activity of infiltrating neutrophils and bacteria, as well as changes in synovial membrane permeability caused by inflammation. Despite this well-known phenomenon in native joints, the role of synovial glucose in periprosthetic joint infection remains underexplored, both as a diagnostic biomarker and as a potential predictor of treatment failure.

We hypothesized that the pathophysiological mechanisms influencing synovial glucose levels in native joint infections may also apply to PJIs, particularly in the acute postoperative setting. Our previous research demonstrated that synovial glucose can be a valuable diagnostic marker in acute postoperative PJI following total knee arthroplasty (TKA) [17]. We compared patients operated under TKA with an arthrocentesis performed in the postoperative period of 90 days, and we obtained a PJI group and a control group finding that synovial glucose decreases in acute postoperative infection. Building on these findings, we aimed to investigate whether synovial glucose—and the serum-to-synovial glucose ratio—could also serve as a prognostic biomarker in this patient population.

The objective of this study was to determine the predictive value of synovial glucose and the serum-to-synovial glucose ratio following DAIR in patients with acute postoperative PJI after TKA.

## 2. Materials and Methods

This was a retrospective prognostic study conducted in patients diagnosed with acute postoperative PJI following TKA. Acute postoperative PJI was defined according to established criteria as any infection occurring within the first 90 days after the index arthroplasty procedure [18]. Diagnosis was confirmed either by positive synovial fluid cultures or by meeting at least three minor criteria from the 2013 Musculoskeletal Infection Society (MSIS) definition [18].

To ensure homogeneity of the cohort and focus on primary joint arthroplasties, we excluded patients who had undergone non-primary TKA procedures (such as revisions or unicompartmental conversions). We retrospectively reviewed all DAIR procedures performed for acute postoperative PJI after TKA between 1 January 2010 and 31 December 2021. Only patients who underwent preoperative joint aspiration with synovial fluid collection were included in this study.

Synovial fluid samples were obtained under sterile conditions and submitted for both microbiological culture and chemical analysis, which in our institution includes routine measurement of synovial glucose. Samples for glucose analysis were collected in tubes without anticoagulant to minimize glycolysis caused by cellular metabolism, which could otherwise result in falsely decreased glucose values. Synovial glucose concentration was determined using an automated biochemical analyzer in the hospital’s central laboratory.

We collected the following variables from medical records: demographic data (age, sex), infecting microorganism, time from surgery to aspiration, and serum laboratory values including C-reactive protein (CRP), glucose, red blood cell count, and white blood cell (WBC) count. Synovial parameters included glucose concentration, total WBC count, and percentage of polymorphonuclear cells (PMN%). Serological and synovial tests were obtained at the same time in the emergency room.

All patients underwent DAIR. This surgery involves thorough debridement of infected tissues while retaining stable implants, exchange of modular components, and high-volume irrigation. Systemic antibiotics are administered based on culture data. In our institution, this surgery is performed by specialized surgeons in PJI and tries to avoid emergent debridement by non-specialized surgeons

Clinical outcomes were evaluated at 3, 12, and 36 months following the DAIR procedure. Treatment failure was defined as the presence of any of the following: death related to PJI, prosthesis removal, chronic suppressive antibiotic therapy, relapse of infection with the same organism, or reinfection with a different microorganism. We also recorded overall mortality and cause of death.

Statistical analysis

To evaluate the prognostic utility of synovial glucose, we first calculated the serum-to-synovial glucose ratio for each patient using the following formula:Ratio = (Serum glucose − Synovial glucose) / Serum glucose 

A multivariable survival analysis was performed using the Cox proportional hazards model to assess the impact of synovial glucose and other covariates on failure risk over time. Hazard ratios (HR) with corresponding 95% confidence intervals were calculated for each explanatory variable. Additionally, Kaplan–Meier survival curves were generated to estimate the survival function, and comparisons between groups were made using the log-rank test.

To determine a clinically relevant cut-off value point for the continuous variables of synovial glucose and the serum-to-synovial glucose ratio in relation to failure-free survival after DAIR, we explored first the relationship between the variables and time-to-event data, and identified an optimal threshold based on survival differences. Once the cut-off was determined, Kaplan–Meier survival curves were compared between groups and stratified patients into high-risk and low-risk groups for treatment failure. These cut-off points were then applied to perform further survival analysis based on group stratification.

All statistical analyses were conducted using Jamovi software (The Jamovi Project, Version 2.3, 2023; Sydney, Australia), with statistical significance set at a two-sided alpha level of 0.05.

## 3. Results

We identified 32 patients who were diagnosed with acute postoperative PJI after TKA who had a synovial joint aspiration (Figure 1). All these patients underwent DAIR procedure and postoperative antibiotic therapy for 6 weeks. The median (IQR) age of the patients in our study was 72.5 (14.8) years. The median (IQR) time interval between the TKA surgery and joint fluid aspiration was 30.5 (26.3) days. In terms of demographic characteristics, 65.6% were female and 62.5% were left knees. Twenty-one patients (65.6%) were classified as ASA II, nine (28.1%) as ASA III, and two (6.3%) as ASA I. Only two patients presented with negative cultures. The majority of infections were monomicrobial (87%). *Staphylococcus aureus* was the most common microorganism (40%), followed by coagulase-negative staphylococci (CoNS) (10%). The remaining 50% consisted of various gram-negative bacteria (GNB), including *Pseudomonas aeruginosa*, *Escherichia coli*, *Serratia marcescens*, *Klebsiella pneumoniae*, *Enterobacter cloacae*, *Morganella morganii*, *Bacteroides fragilis*, and *Proteus mirabilis*.

Ten patients (31.3%) experienced failure following DAIR. Of these, half encountered failure within the first 3 months post-DAIR, three additional patients within the first year after debridement, and the remaining two after three years. Table 1 shows differences between failure and success. The multivariate survival analysis on gender, cultures, level of CRP, WBC count, PMN%, or synovial glucose could not find any variable associated to failure (Figure 2).

For synovial glucose, we obtained a cut-off value of 44 mg/dL. Patients below this threshold were classified as high risk of failure, 24 patients were allocated in this high-risk failure group, and 8 in the low-risk failure group. We observed 10 failures in the first group and none in the low-risk failure group (*p* = 0.036) (Figure 3a).

For the serum-to-synovial glucose ratio, we obtained a cut-off value of 50%. Patients above this threshold were classified as high risk of failure, 26 patients were allocated in this high-risk failure group, and 6 in the low-risk failure group. We observed 10 failures in the first group and none in the low-risk failure group (*p* = 0.084) (Figure 3b).

Five of the ten failed patients occurred during the first 3 months after DAIR. Patients with early failure presented median glucose levels of 4 mg/dL and a serum to synovial glucose ratio of 97.3%; patients with late failure presented median glucose levels of 32 mg/dL and serum to synovial glucose levels of 66.9% (*p* = 0.094 and *p* = 0.056, respectively) (Table 2).

## 4. Discussion

DAIR is widely considered a preferred surgical strategy for managing acute periprosthetic joint infection (PJI) due to its relative simplicity and lower morbidity compared to revision surgery. However, the outcomes of DAIR remain highly variable, with reported failure rates ranging between 20% and 70% depending on patient selection, surgical technique, and timing of intervention [7,8,9,10,11]. This variability has spurred considerable interest in identifying preoperative predictors of treatment failure to better guide clinical decision-making and avoid ineffective interventions.

Previous studies have identified multiple risk factors for DAIR failure, including host-related characteristics (such as comorbidities and immune status) [6,12,13], factors related to the index surgery (including prosthesis type and surgical complexity) [19,20], and most importantly, the timing and adequacy of debridement [21,22,23]. The quality of the surgical procedure itself and the burden of infection at the time of DAIR are also essential contributors to the outcome. Furthermore, analytical biomarkers have gained attention for their prognostic value, especially serum-based markers.

Several predictive tools, such as the KLIC score and the CRIME80 score, have emphasized the utility of preoperative factors—primarily serological parameters—in estimating the likelihood of DAIR success [12,13]. Shohat et al. [19] expanded on this by proposing a multifactorial model incorporating 10 variables, identifying C-reactive protein (CRP) as the strongest predictor of treatment failure. Despite these advancements, most existing scores either do not incorporate synovial biomarkers or fail to highlight their significance. In recent years, machine learning approaches have also been applied to improve predictive performance, but these too have predominantly relied on systemic laboratory values [19,24].

To the best of our knowledge, this is the first study to specifically investigate the prognostic role of synovial glucose and the serum-to-synovial glucose ratio in predicting DAIR failure for acute postoperative PJI. Our results suggest that these synovial parameters may offer valuable prognostic information beyond traditional serum biomarkers. We identified a synovial glucose threshold of 44 mg/dL and a serum-to-synovial glucose ratio cutoff of 50% as potential discriminators of treatment outcome. Patients falling below the synovial glucose threshold or above the glucose ratio cutoff were significantly more likely to experience failure, while those in the low-risk group showed no treatment failures. These findings support the hypothesis that synovial glucose metabolism reflects local inflammatory activity and may be more directly related to the infectious burden within the joint.

Interestingly, we also observed a temporal pattern in treatment failure. Approximately half of the failures occurred within the first 3 months following DAIR. Although this early failure group did not reach statistical significance when compared to late failures, there was a clear trend toward more extreme synovial glucose alterations, with notably lower synovial glucose levels (median 4 mg/dL) and higher serum-to-synovial glucose ratios (median 97.3%). This could reflect a more aggressive inflammatory response or insufficient infection control at the time of debridement.

The overall failure rate in our cohort was 31.3%, which is consistent with the previous reports in the literature [10,25,26,27]. However, in contrast to other studies, we did not identify additional variables—such as CRP, patient sex, infecting organism, or synovial WBC count—as independent predictors of failure. This could be attributed to sample size limitations or potential differences in the timing and technique of DAIR procedures in our institution.

Our study has several limitations that must be acknowledged. First, the retrospective design inherently carries the risk of selection bias and incomplete data. Notably, a considerable number of patients were excluded due to missing synovial fluid data, which could have introduced a selection bias toward more thoroughly investigated or complex cases. Second, this study was conducted at a single center, which, while enhancing internal consistency in clinical practice and laboratory methods, may reduce external validity. The relatively small sample size further limits the generalizability of our findings and precluded the inclusion of additional variables in multivariate models.

Despite these limitations, this study introduces a novel and easily obtainable synovial biomarker that may enhance preoperative risk stratification in patients undergoing DAIR. Synovial glucose testing is widely available, inexpensive, and already part of routine synovial analysis in many institutions. Incorporating synovial glucose measurements into existing prediction models could improve their accuracy and clinical utility.

In conclusion, our findings suggest that low synovial glucose levels and elevated serum-to-synovial glucose ratios are associated with an increased risk of failure following DAIR for acute postoperative PJI. Patients with extreme synovial glucose alterations were more likely to experience early treatment failure. These results highlight the potential utility of synovial glucose as a prognostic biomarker in clinical practice. However, larger prospective and multicenter studies are warranted to validate these findings and determine their role in improving patient selection and outcomes in the treatment of acute PJI.

## Figures and Tables

**Figure 1 jcm-14-02841-f001:**
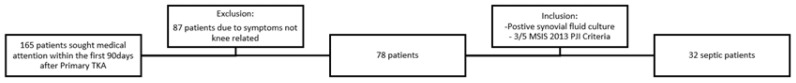
Patient inclusion flowchart.

**Figure 2 jcm-14-02841-f002:**
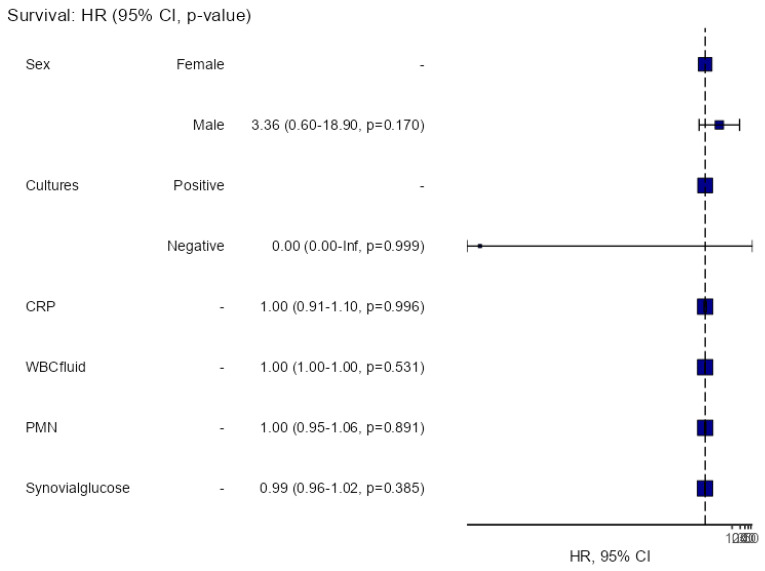
Multivariate survival analysis.

**Figure 3 jcm-14-02841-f003:**
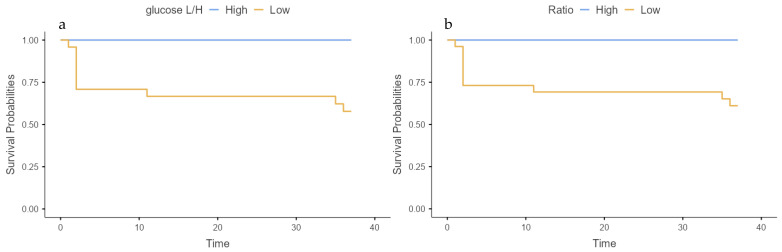
Survival curve: (**a**) synovial glucose and (**b**) serum-to-synovial glucose ratio.

**Table 1 jcm-14-02841-t001:** Demographic data and serological differences in failure and success.

	Failure (n = 10, 31.3%)	Succes (n = 22, 68.7%)	*p* Value
Age	71.5 years (16.5)	72.5 years (13.5)	0.791
Sex			
Male	5 (50%)	16 (72.7%)	0.213
Female	5 (50%)	6 (23.3%)	
Laterality			
Right	5 (50%)	7 (31.8%)	0.270
Left	5 (50%)	15 (68.2%)	
Time lapse	27.5 days (18.8)	34 days (26.8)	0.597
Cultures			
Positive	10 (100%)	20 (90.1%)	0.998
Negative	0	2 (9.9%)	
CRP	16.2 mg/dL (21.5)	8.6 mg/dL (7.25)	0.366
WBC	4910 cells (18,891)	8550 cells (31,521)	0.730
PMN	90% (8.25)	90% (27)	0.628
Synovial glucose	14 mg/dL (28.3)	25 mg/dL (56.8)	0.328
Serum-to-synovial-glucose ratio	87.8% (29.3)	75.2% (41.9)	0.509

CRP: C reactive protein; WBC: white blood cells; PMN%: polymorphonuclear percentage.

**Table 2 jcm-14-02841-t002:** Synovial glucose and serum-to-synovial glucose ratio in failure.

	Early Failure (<3 Months) (n = 5)	Late Failure (>3 Months) (n = 5)	*p* Value
Synovial glucose	4 mg/dL	32 mg/dL	0.094
Serum-to-synovial glucose ratio	97.3%	66.9%	0.056

## Data Availability

The raw data supporting the conclusions of this article will be made available by the authors on request.

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
