# Peer review of "Synovial Glucose and Serum-to-Synovial Glucose Predict Failure After Acute Postoperative Infection in Total Knee Arthroplasty"

_jcm, 2025, doi:10.3390/jcm14082841_

Round 1

Reviewer 1 Report

Comments and Suggestions for Authors

The article explores the relationship between synovial glucose levels and the synovial-to-serum glucose ratio as prognostic markers of failure following DAIR (Debridement, Antibiotics, and Implant Retention) in the setting of acute periprosthetic joint infection after total knee arthroplasty. This represents a novel perspective on the clinical relevance of a classic infection marker such as glucose consumption. The main limitation identified is the sample size.

Areas for Improvement

  • Title: The current title, “Synovial glucose and serum-to-synovial glucose predict failure after acute postoperative total knee arthroplasty”, does not explicitly mention periprosthetic joint infection. I suggest the following revision: “Synovial glucose and serum-to-synovial glucose predict failure after acute postoperative infection over total knee arthroplasty.”

  • Hypothesis Explanation: The hypothesis should be better justified. Under normal conditions, synovial glucose levels are similar to those in blood; however, they may decrease in the presence of inflammation or infection. At one month post–total knee arthroplasty, inflammation may still be present. Therefore, could this be inflammation without infection? The validity of the marker could be better assessed by including a control group of non-infected total knee arthroplasties and analyzing their glucose ratios.

  • Methodology: Consider including a flowchart to outline patient selection criteria. Please include exclusion criteria as well.

  • Control Group: Consider adding a control group to evaluate how the glucose ratio behaves during the immediate postoperative period of total knee arthroplasty.

  • Sample Size: Include a sample size calculation or at least the statistical power of the sample used for the analysis of the studied variables.

  • Cut-off Determination: Detail the strategy or methodology used to define the applied cut-off values.

Formatting Issues

  • References: In the main text, references begin at line 46, starting with references 10 and 11 in the Introduction. Please revise the numerical order and placement of in-text citations.

  • Table 2: Indicate the number of patients in the early and late failure groups. I suggest adding this information directly into the column headers of the table.

  • Line 261: The ethics approval date is missing. Please provide this information.

Author Response

Thank you for your time and comments that fo sure will help improve our research

  • Title: The current title, “Synovial glucose and serum-to-synovial glucose predict failure after acute postoperative total knee arthroplasty”, does not explicitly mention periprosthetic joint infection. I suggest the following revision: “Synovial glucose and serum-to-synovial glucose predict failure after acute postoperative infection over total knee arthroplasty.”

Thank you for your comment. Tittle has been changed adding the infecction concept.

  • Hypothesis Explanation: The hypothesis should be better justified. Under normal conditions, synovial glucose levels are similar to those in blood; however, they may decrease in the presence of inflammation or infection. At one month post–total knee arthroplasty, inflammation may still be present. Therefore, could this be inflammation without infection? The validity of the marker could be better assessed by including a control group of non-infected total knee arthroplasties and analyzing their glucose ratios.

Thank you for your comment. We added “ We compared patients operated of TKA with an arthrocentesis performed in the postoperative period of 90 days, we obtained a PJI group and a control group finding that synovial glucose decreases in acute postoperative infection” to the introdcction so the reader can understand the background of this research.

  • Methodology: Consider including a flowchart to outline patient selection criteria. Please include exclusion criteria as well.

Thank you for your comment. A Flow chart was included as suggested as Figure 1 and changed all figures number accordingly

  • Control Group: Consider adding a control group to evaluate how the glucose ratio behaves during the immediate postoperative period of total knee arthroplasty.

Thank you for your comment. Differences in Synovial glucose as a diagnostic tool has already been studied in a previous sutdy of our team. In this study we describe the behaviour of Synovial glucose and the ratio in PJI and control group.

Sabater-Martos M, Garcia O, Boadas L, Morata L, Soriano A, Martínez-Pastor JC. Synovial glucose and serum-to-synovial-glucose ratio perform better than other biomarkers for the diagnosis of acute postoperative prosthetic knee infection. J Bone Jt Infect. 2025 Mar 4;10(2):41–9.

  • Sample Size: Include a sample size calculation or at least the statistical power of the sample used for the analysis of the studied variables.

Thank you for your comment. It’s true that a sample size calculation would add statistical power to our analysis. However, in PJI sample size calculations are impossible to achieve. For example, this article required a sample size of 2965 patients. Therefore, prospective studies are not feasible.

  • Cut-off Determination: Detail the strategy or methodology used to define the applied cut-off values.

Thank you for your comment. We added the following paragraph to define the methodology for cut-off values calculation.

“To determine a clinically relevant cut-off value point for the continuous variables synovial glucose and the serum-to-synovial glucose ratio in relation to failure-free survival after DAIR we explored first the relationship between the variables and time-to -event data, and identified an optimal threshold based on survival differences. Once the cut-off was determined, Kaplan-Meier survival curves were compared between groups and stratified patients into high-risk and low-risk groups for treatment failure. These cut-off points were then applied to perform further survival analysis based on group stratification.”

Formatting Issues

  • References: In the main text, references begin at line 46, starting with references 10 and 11 in the Introduction. Please revise the numerical order and placement of in-text citations.

Thank you for your comment. This mismatch has been amended. Bibliography has been corrected

  • Table 2: Indicate the number of patients in the early and late failure groups. I suggest adding this information directly into the column headers of the table.

Thank you for your comment. We have added the number of patients in table 2.

  • Line 261: The ethics approval date is missing. Please provide this information.

The date has been added.

Reviewer 2 Report

Comments and Suggestions for Authors

Synopsis:

Thank you for submitting the manuscript entitled "Synovial glucose and serum-to-synovial glucose ratio predict failure after acute postoperative TKA" to JCM.
Thirty-two cases undergoing DAIR for acute PJI were analysed. Synovial aspirates were taken  and serum and synovial glucose levels were compared. The ratio of serum to synovial glucose was calculated. The aim of the study was to identify predictors of unsuccessful outcomes after DAIR. The authors concluded: "Low synovial glucose levels and high serum-to-synovial glucose ratios are predictive of unsuccessful outcomes after DAIR procedures. Patients with lower synovial concentrations experienced early treatment failure.

Abstract:
The abstract is well written and summarizes the study well.

Title:

Clear, adequate length 14 words

Introduction/Materials

The Introduction section is properly written, Materials are well described.

Results

Overall clear.

Patients who developed PJI within 90 days of the index surgery were included. What was the interval in the cohort? Is there a trend in synovial glucose concentration or serum/synovial Glc ratio over the 90-day period? Does a longer period correlate with a higher risk of failure?

Discussion:

The discussion is well written. Limitations were also discussed.

In summary, the manuscript is of a satisfactory standard.

Author Response

Dear reviewer, thank you for your time and comments that will help imrpove this article. We added the information on time lapse requested .

Synopsis:

Thank you for submitting the manuscript entitled "Synovial glucose and serum-to-synovial glucose ratio predict failure after acute postoperative TKA" to JCM.
Thirty-two cases undergoing DAIR for acute PJI were analysed. Synovial aspirates were taken  and serum and synovial glucose levels were compared. The ratio of serum to synovial glucose was calculated. The aim of the study was to identify predictors of unsuccessful outcomes after DAIR. The authors concluded: "Low synovial glucose levels and high serum-to-synovial glucose ratios are predictive of unsuccessful outcomes after DAIR procedures. Patients with lower synovial concentrations experienced early treatment failure.

Abstract:
The abstract is well written and summarizes the study well.

Title:

Clear, adequate length 14 words

Introduction/Materials

The Introduction section is properly written, Materials are well described.

Results

Overall clear.

Patients who developed PJI within 90 days of the index surgery were included. What was the interval in the cohort? Is there a trend in synovial glucose concentration or serum/synovial Glc ratio over the 90-day period? Does a longer period correlate with a higher risk of failure?

Thank you for your comment. We added this information on time lapse between groups in table 1

Discussion:

The discussion is well written. Limitations were also discussed.

In summary, the manuscript is of a satisfactory standard.

Reviewer 3 Report

Comments and Suggestions for Authors

Interesting evaluation, at least from the practitioners point of view (reviewer is orthopaedic surgeon). But still it should be widened in many points.

I miss following information:

  • what determines the glucose levels in "healthy" joints (introduction)?
  • which mechanism is determining the decrease of synovial glucose levels in joint infection (introduction) ?
  • is there any literature supporting the relation of glucose synovial level and the "infection activity" (introduction or discussion)?
  • is there any link between diabetes and synovial glucose (introduction or discussion)?

When was the serum glucose collected? And why not glycated hemoglobin (especially for this study)

There are relevant limitations : small sample size, patients examined and treated with single center, Authors mention it. 

Authors do not evaluate patients in terms of the predisposing factors (general and local) that could predispose to joint infections, subsequently to increased risk of DAIR failure.

DAIR procedure should be described in detail (technically). 

Author Response

Thank you for your comments that for sure will help improve this reasearch

I miss following information:

  • what determines the glucose levels in "healthy" joints (introduction)?

Thank you for your comment. This information has been added to the introduction. In a previous study we demonstrated the normal glucose level in healthy operated patients.

  • which mechanism is determining the decrease of synovial glucose levels in joint infection (introduction) ?

Line 59-62 in introduction explains how the synovial glucose levels can decrease in a native septic arthritis. As exposed, we hypothesized that the pathophysiological mechanisms influencing synovial glucose levels in native joint infections may also apply to PJIs

  • is there any literature supporting the relation of glucose synovial level and the "infection activity" (introduction or discussion)?

Thank you for your comment. Information on synovial glucose and glucose ratio is referenced in the introduction number 15 and 16.

  • is there any link between diabetes and synovial glucose (introduction or discussion)?

Under normal conditions (without sepsis or inflamation), the concentration of glucose in synovial fluid is very similar to that of serum glucose. Therefore, glucose ratio provides better information on glucose drop.  In our article we provide both parameters.

  • When was the serum glucose collected? And why not glycated hemoglobin (especially for this study).

Thank you for your comment. We added the information on when was the sample collected. This study is not on diabetic patients, for this reason glycated Hb was not considered.

  • There are relevant limitations: small sample size, patients examined and treated with single center, Authors mention it. 
  • Authors do not evaluate patients in terms of the predisposing factors (general and local) that could predispose to joint infections, subsequently to increased risk of DAIR failure.

Thank you for your comment. We evaluated factors described in KLIC score that is the only score known to predict failure on acute postoperative PJIs.

  • DAIR procedure should be described in detail (technically). 

Thank you for your comment. We added this information in material and methods.

Round 2

Reviewer 1 Report

Comments and Suggestions for Authors

The authors have responded satisfactorily, making the suggested improvements.

Reviewer 3 Report

Comments and Suggestions for Authors

I am convinced with author response.

I suggest only that all Authors' Responses to Reviewer's Comments should be incorporated to the text.